# The Mental Health of Employees with Job Loss and Income Loss during the COVID-19 Pandemic: The Mediating Role of Perceived Financial Stress

**DOI:** 10.3390/ijerph19063158

**Published:** 2022-03-08

**Authors:** Carlota de Miquel, Joan Domènech-Abella, Mireia Felez-Nobrega, Paula Cristóbal-Narváez, Philippe Mortier, Gemma Vilagut, Jordi Alonso, Beatriz Olaya, Josep Maria Haro

**Affiliations:** 1Research, Innovation and Teaching Unit, Parc Sanitari Sant Joan de Déu, 08830 Sant Boi de Llobregat, Spain; carlota.demiquel@sjd.es (C.d.M.); joan.domenech@sjd.es (J.D.-A.); mireia.felez@sjd.es (M.F.-N.); paula.cristobal@sjd.es (P.C.-N.); josepmaria.haro@sjd.es (J.M.H.); 2Centro de Investigación Biomédica en Red de Salud Mental (CIBERSAM), 28029 Madrid, Spain; 3Health Services Research Unit, Hospital del Mar Medical Research Institute (IMIM), 08003 Barcelona, Spain; pmortier@imim.es (P.M.); gvilagut@imim.es (G.V.); jalonso@imim.es (J.A.); 4CIBER Epidemiología y Salud Pública (CIBERESP), 28029 Madrid, Spain; 5Department of Medicine and Life Sciences, Universitat Pompeu Fabra, 08003 Barcelona, Spain

**Keywords:** mental health, COVID-19, income loss, job loss, financial stress, mediation

## Abstract

The COVID-19 outbreak, which was followed by home confinement, is expected to have had profound negative impact on the mental health of people. Associated factors, such as losing jobs and income, can be expected to lead to an increased risk of suffering from psychopathological problems. Therefore, this study was aimed at researching the associations of job and income loss with mental health, as well as the possible mediating role of perceived financial stress during the COVID-19 outbreak. The sample included 2381 Spanish workers who were interviewed right after the first COVID-19 lockdown. Measures were taken for generalized anxiety disorder, panic attacks, depression, post-traumatic stress disorder, substance abuse, suicidal thoughts and behaviors, working conditions, sociodemographic variables, and perceived financial stress. Logistic regression models were calculated with psychological variables as outcomes, and with job loss and income loss as predictors. Mediation analyses were performed by adding the financial threat as a mediator. Nineteen point six percent and 33.9% of participants reported having lost their jobs and incomes due to the pandemic, respectively. Only income loss was related to a higher risk of suffering from depression and panic attacks. When adding financial stress as a mediator, the indirect effects of job and income loss on the mental health measures were found to be significant, therefore indicating mediation. These findings pinpoint the vulnerability of this population, and highlight the need for interventional and preventive programs targeting mental health in economic crisis scenarios, such as the current one. They also highlight the importance of implementing social and income policies during the COVID-19 pandemic to prevent mental health problems.

## 1. Introduction

In December 2019, a novel coronavirus was identified in a wet market in Wuhan, China [1]. Since the very beginning, the COVID-19 pandemic has had a significant negative impact on the psychological well-being of people [2]. This virus causes an acute respiratory syndrome (COVID-19) that rapidly spread around the world. The World Health Organization (WHO) officially declared COVID-19 as a pandemic in March 2020 [3]. In China, Spain, Italy, Iran, the USA, Turkey, Denmark, and Nepal, increased rates were found for anxiety (6.33% to 50.9%), depression (14.6% to 48.3%), post-traumatic stress disorder (PTSD; 7% to 53.8%), psychological distress (34.43% to 38%), and stress (8.1% to 81.9%) [4]. Such increased rates could have been caused by different factors in the context of the pandemic, such as job unemployment or a loss of income. In 2020, there was a loss of 8.8% of global working hours relative to the fourth quarter of 2019, which is equivalent to 255 million full-time jobs [5]. This represents an approximately four-times greater loss than the one observed during the global financial crisis of 2009 [5]. Specifically, in Spain, unemployment increased by 22.9% from December 2019 to December 2020, with the highest increase observed in March (9.31%, as compared to February) and April 2020 (7.97%, as compared to March) [6]. Many workers were also temporarily laid off. By the end of 2020, the number of temporarily laid off workers in Spain was 755,613 [7], which, in turn, could result in a lower income.

Since, in this situation, the source of such an economic impact is mostly exogenous to the individual and is mainly due to the COVID-19 pandemic, this offers a unique opportunity to analyze the implications of job and income loss on mental health. Previously, the relationship between unemployment and mental health was more difficult to establish, as there were methodological problems associated with causality. Although there is a large amount of literature which has focused on the effects of employment on mental health [8,9,10], it has been difficult to establish whether poor mental health affected losing jobs, or if it was the other way around [11]. According to the International Labour Organization’s press release from 18 March 2020, a reported 5.3 million jobs will be lost due to the COVID-19 pandemic in the low scenario, and 24.7 million in the high scenario [12]. In terms of mental health, in the high scenario, this would translate into an increase in suicides of about 9570 per year. In the low scenario, unemployment would be associated with an increase of about 2135 suicides.

From the experience of past economic crises, job insecurity conditions, such as the ones workers are experiencing during the pandemic, is thought to lead to poorer mental health outcomes. Three independent systematic reviews concluded that people who became unemployed during a recession and/or experienced financial crises were more likely to suffer from poorer health and increased stress, depression, mental hardship, anxiety, and suicidal behaviors [13,14,15]. Social security programs have the power to buffer the adverse health impacts that such financial crises can cause. Indeed, a review of studies conducted after the recession of 2008 found that unemployed individuals who resided in countries offering better unemployment benefits reported increased physical and mental health due to greater financial security [16]. In Spain, poor mental health increased from 2005 to 2010, corresponding with a time where the economy crashed [17]. A systematic review also identified several vulnerable groups that experience negative mental health outcomes during an economic crisis, such as the unemployed, people with debt, or people facing financial difficulties [13]. In fact, some studies suggest that every percentage point increase in unemployment translates into a rise of 0.79% in suicides in people under the age of 65 [18]. Due to the increment of unemployment rates and a decrease in income during the COVID-19 pandemic, an increased risk for mental health problems is expected.

Several studies have investigated the impact of work-related risk factors on mental health problems during the COVID-19 pandemic. A population-based study carried out in three different countries (USA, United Kingdom, and Israel) found that the level of anxiety related to the economic situation was equal to the level of anxiety related to health during the pandemic [19]. A recent narrative review showed that the factors that seem to be relevant for the mental health of workers during the pandemic are being a high-risk professional, work-related stress, a lack of job support, belonging to higher risk populations, and being a young adult worker [20]. This review highlighted that job insecurity, adverse employment environments, long periods of quarantine and isolation, work rights exploitation, and uncertainty about the future are strongly associated with worse mental health outcomes. Similarly, a study conducted with USA young adults during the COVID-19 pandemic showed that poor mental health outcomes were 2- to 6-times higher for young adults who had either experienced or anticipated employment loss [21]. Indeed, a study conducted in the USA found that individuals who had lost their jobs reported higher symptoms of depression, anxiety, and stress [22]. The same was found to be true in a study conducted in South Africa [23]. Additionally, in other studies conducted during this period, financial loss has also been related to higher symptoms of mental health problems, such as anxiety or depression [24,25]. Overall, these results highlight the importance of work-related variables, and, specifically, job and income loss, in the wellbeing of people during the COVID-19 pandemic.

However, the subjective experiences when losing a job or income may vary, and, thus, the impact of job and income loss on mental health might be different for each individual. The individual perception of a threat often has more to do with the subjective perceived danger than with the reality itself [26]. In the context of economic hardship, recent evidence has identified a possible mediator of its relationship with mental health, namely, perceived financial stress [27,28]. Financial threat is defined as a self-reported fearful-anxious uncertainty regarding the current and future financial situation [29]. Such financial threat has been found to be at its highest in individuals who have experienced economic hardship [27]. A situation of job loss or income loss might lead to this perceived financial threat and stress. Financial threat has been, in turn, positively associated with depression, anxiety, mood disturbances, burnout, and suicidal ideation [28]. In fact, when controlling for perceived financial stress, the relationship between economic hardship and mental health seems to diminish [27]. Indeed, several studies conducted during the COVID-19 pandemic encountered the effect of perceived financial stress on mental health [30,31,32]. As an example, a study regarding the effect of COVID-19 on mental health in the Australian population showed that the financial distress due to the pandemic was an important correlate of worse mental health, rather than job loss *per se* [30]. Additionally, another recent study conducted during the COVID-19 pandemic in the USA found that the relationship between job insecurity and anxiety symptoms was mediated by financial concern, again reflecting the importance of the subjective experience of job and income loss [31]. If perceived financial stress mediates the impact of job loss and income loss on mental health during the COVID-19 pandemic, this could help in the design and adoption of interventions for employees to improve their coping strategies and decrease their level of stress, which, in turn, might result in an improvement in their mental health. Some strategies have been proposed to reduce the negative psychological state of workers facing the consequences of COVID-19. For example, companies could arrange professional counselors for advice and consultation, or they could provide employees with the ability to control and arrange their work setting in order to help them reduce the negative pressure caused by job insecurity [33].

As different countries and regions adopted distinct measures to stop the COVID-19 transmission, the impact of the economic consequences in one country might not reflect the consequences in another. Therefore, the aim of this study was to assess the association of job and income loss with mental health problems in Spanish workers right after the first COVID-19 lockdown. Additionally, we aimed to explore the potential mediating role of stress provoked by the financial situation in this relationship (i.e., perceived financial stress). It was expected that job and income loss would be associated with increased mental health problems (such as depression, anxiety, PTSD, panic attacks, substance abuse, and suicidal thoughts and behaviors (STBs)). Moreover, perceived financial stress was expected to mediate these relationships. Specifically, it was expected that job and income loss would lead to an increase in stress regarding the financial situation, which, in turn, would result in a higher risk of mental health problems. Having a better understanding on how the economic consequences of the COVID-19 pandemic have affected the Spanish population could guide employers and policymakers to adopt measures to address the issue.

## 2. Materials and Methods

### 2.1. Sample and Study Design

We analyzed data from a cross-sectional survey conducted in a representative sample of non-institutionalized adults in Spain as part of the MIND/COVID project [34]. The target population of the survey included people who were both aged 18 years or older and had no language barriers. Computer-assisted interviews (CATI) were carried out by a bureau of professional interviewers during June 2020. The sample was drawn through a dual-frame random digit dialing (RDD) telephone survey, and it included both landlines and mobile telephones. The distribution of the interviews was planned according to quotas proportional to the Spanish population in terms of age groups, sex, and the region of residence [35].

A total of 138,656 numbers were sampled, with a final split of 71% mobile telephones and 29% landline telephones. Of them, 45,002 were classified as non-eligible (i.e., 43,120 had non-existing numbers, 984 were numbers of enterprises, 444 were numbers of persons with Spanish language barriers, 268 were fax numbers, 186 numbers belonged to the quota that was already completed), and 72,428 had unknown eligibility (i.e., no contact was made after the seven attempted calls), resulting in a 19.7% cooperation rate (i.e., the proportion of all cases interviewed of all eligible units ever contacted). Finally, 3500 people were interviewed during the COVID-19 lockdown in Spain, from which only 2381 belonged to the active population (see Figure 1). Interviews included general demographic questions and questions related to social networks and living situations, socioeconomic factors, the use of health resources, mental health, general health, and wellbeing. For mental health, depression, anxiety, PTSD symptoms, panic attacks, substance abuse, and suicidal thoughts and behaviors were asked by means of validated questionnaires. Ethical approval was provided by Fundació Sant Joan de Déu, Barcelona, Spain (PIC 86−20) and by the Parc de Salut Mar Clinical Research Ethics Committee (protocol 2020/9203/I). Oral consent was obtained once the selected participant was fully informed about the objectives and procedures of the study to proceed with the interview.

### 2.2. Sociodemographic Measures

Sociodemographic variables included age, gender, marital status (single, divorced or legally separated, widowed, and married), and educational level (primary school, secondary school, high school, professional training—intermediate degree, professional training—higher degree, bachelor’s degree, and postgraduate degree).

### 2.3. Mental Health Measures

The eight-item Patient Health Questionnaire depression scale (PHQ-8) [36] was used to measure depression symptoms in the previous two weeks. The scores for the PHQ-8 range from 0 to 24, where higher numbers indicate a higher number of depressive symptoms. The cut-off point of 10 was used to indicate current depression. The PHQ-8 has been found to show a high reliability (>0.8) [37].

The General Anxiety Disorder Scale (GAD-7) [38] was used to measure anxiety symptoms during the previous two weeks. This scale is considered to have acceptable sensitivity and specificity for detecting anxiety [39]. The GAD-7 scores range from 0 to 21, where higher scores represent greater anxiety symptoms. A cut-off of 10 was set to indicate current GAD.

PTSD symptoms for the previous 30 days were measured by means of the four items of the PTSD Checklist for the DSM-5 (PCL-5) [40,41]. The use of four items generates diagnoses that closely parallel those of the full PCL-5, which results in a well-suited instrument for PTSD screening [41]. Scores range from 0 to 16, where higher scores indicate the presence of PTSD symptoms. The cut-off point for current PTSD was set at 7 points.

An item from the World Mental Health-International College Student (WMH-ICS) [42] survey was used to assess the number of panic attacks in the 30 days prior to the interview. A dichotomous variable was created to indicate the presence of panic attacks.

The CAGE-AID questionnaire [43] was used to evaluate substance use disorders (SUD). This questionnaire has proven to be useful when diagnosing both alcoholism and SUD [44]. A score of two points was set as a cut-off to indicate current SUD.

Finally, having had any STBsin the previous 30 days was determined by means of a modified version of selected items from the Columbia Suicide Severity Rating Scale (C-SSRS, [45]). These items consisted of questions with dichotomous (yes/no) answers and assessed passive suicidal ideation, active suicidal ideation, suicide plans, and suicide attempts. All responses were combined, and a general dichotomous variable was created measuring if any of the questions were answered with a “yes”.

### 2.4. Work-Related Measures

To assess whether the individual had suffered from job loss, a temporary lay-off, or from income loss, two dichotomous questions (Y/N) were asked: “Are you unemployed or temporarily laid off (in Spanish: Expediente de Regulación Temporal de empleo, i.e., ERTE) due to the coronavirus pandemic?” and “Did you experience a significant loss of personal or family economic income due to the coronavirus pandemic?” Therefore, the variable “job loss” entails both people who became unemployed as a consequence of the COVID-19 pandemic, as well as the people that suffered from a temporary lay-off. 

Perceived financial stress was assessed by an adapted version of the Peri Life Events Scale [46] by asking “How much stress does each of the following aspects of your life cause you? Your financial situation”. The answer followed a 5-point Likert scale from None (1) to Very Intense (5), with a higher score representing higher perceived financial stress. 

### 2.5. Statistical Analyses

Data was weighted with post-stratification weights to restore the distribution of the adult general population of Spain according to age groups, sex, and geographic areas, to compensate for survey non-responses and to ensure the representativeness of the sample. Descriptive statistics for both the weighted and unweighted datasets were estimated for sociodemographic, mental health, and work-related variables. Additionally, an independence test was performed between the two main explanatory variables, namely, job loss and income loss.

First, two unadjusted logistic regression models were estimated with the mental health conditions (depression, anxiety, PTSD, panic attacks, SUB, and STB) as dependent variables. Job loss was an independent variable in one of the models and income loss in the other model. The odds ratios (OR) and 95% confident intervals (CI) were calculated. Then, the adjusted logistic regressions were estimated, with the same mental health conditions as outcomes and job loss and income loss as the main explanatory variables. Gender, age, education level, and marital status were covariates. Two separate models were created, one for each of the outcome variables. Interactions between job or income loss, and other variables, such as gender, age, education, or marital status, were further explored in the adjusted models.

To assess the mediation of perceived financial stress in the association between job loss and income loss with each mental health condition, mediational analyses were performed [47]. Mental health outcomes (i.e., depression, GAD, PTSD, panic attacks, SUD, and STB) were added as dependent variables, and job loss or income loss were added as independent variables with perceived financial stress as the mediator. We first determined the association between the independent variable and the mediator and whether the mediator was associated with the dependent variable. Second, we tested whether the independent variable was related to the dependent variable when controlling for the mediator. This procedure breaks down the total effect of a variable into direct and indirect (i.e., mediational) effects. The direct effect represents the effect of the independent variable on the dependent one when including the mediator. The indirect effect, in turn, refers to the effect that the independent variable has on the dependent variable that is explained by the mediator [48,49]. Models were adjusted for confounding factors, such as gender, age, educational level, and marital status. In the analyses, a significant indirect effect indicates mediation, irrespective of whether the total and the direct effects are significant [50]. 

Bootstrap techniques with 5000 simulations were used by means of the mediation package from *R* [51,52]. The principal advantage of bootstrapping is that this method makes no assumptions regarding the sampling weight, with at least 1000 simulations needed, with 5000 as the recommended amount [47,48]. Beta coefficients with 95% confidence intervals and *p*-values for direct, indirect, and total effects in each case are displayed. 

## 3. Results

### 3.1. Sample Sociodemographics

The total number of participants in the survey was 3500. However, as we aimed to analyze the impact of job or income loss, for the analysis, only the active population was considered (*n* = 2381). Active population includes individuals who are either currently employed or currently unemployed with or without receiving benefits or subsidies. A summary of the weighted sociodemographic, mental health, and work-related information can be found in Table 1. Eleven point six percent of the respondents presented current depression, 11.4% presented a GAD, 10.6% presented PTSD, 3.6% presented SUD, 10.4% reported at least one panic attack in the last 30 days, and 3.8% reported having STB in the last 30 days. A total of 26.9% of respondents reported having lost their job or being temporarily laid off due to the COVID-19 pandemic, and 42.5% had suffered income loss as a result of the pandemic. Job loss and income loss were found to be significantly related to one another (*χ^2^* = 222.02, *df* = 1, *p* < 0.001).

### 3.2. Regression Analyses

Unadjusted logistic regression models and their respective ORs were calculated for the associations between job loss and income loss with depression, GAD, PTSD, SUD, STB, and panic attacks in two separated models (see Table 2). The unadjusted ORs of having current depression were 48% higher for people who suffered from job loss (OR = 1.48, 95% CI = 1.12–1.95), and 30% higher for those who suffered from income loss (OR = 1.30, 95% CI = 1.00–1.69). Job loss (OR = 1.35, 95% CI = 1.01–1.81) and income loss (OR = 1.37, 95% CI = 1.04–1.79) significantly increased the risk of having PTSD. For STB, only losing job was found to be a significant risk factor (OR = 2.11, 95% CI = 1.36–3.28). For panic attacks, only people who suffered income loss were found to be at a higher risk than those who did not (OR = 1.49, 95% CI = 1.13–1.96). Neither job loss nor income loss were found to be significantly associated with GAD or SUD. 

In the adjusted logistic regression models (Table 3), no increased risk for depression, GAD, PTSD, or SUD was observed for individuals who had experienced job loss or income loss. People who experienced job loss were at a higher risk of suffering from STB (OR = 1.67, 95% CI = 1.06–2.63), compared to those who did not lose their job. Additionally, for people who experienced income loss, the risk of having experienced at least one panic attack was 39% higher (OR = 1.40, 95% CI = 1.09–1.80), than those who did not experience income loss.

### 3.3. Mediation Analyses

As both job and income losses were significantly associated with financial stress (*β* = 0.67, *p* < 0.001 and *β* = 0.83, *p* < 0.001, respectively), we proceeded with the mediation analyses. Figure 2 shows the hypothesized mediational analyses for job loss and perceived stress about one’s financial situation. The direct arrow from job loss to mental health outcomes represents the direct effect. The path from job loss to mental health outcomes through perceived financial stress represents the indirect effect. The total effect is represented by the sum of both direct and indirect effects. The results for these effects are displayed in Table 4. Perceived financial stress significantly mediated the relationship between job loss and all mental health conditions, such as depression (*p* < 0.001), GAD (*p* < 0.001), PTSD (*p* < 0.001), panic attacks (*p* < 0.001), SUD (*p* = 0.01), and STB (*p* = 0.003). The direct effect of job loss on depression, panic attacks, SUD, and STB was non-significant, indicating that the effect of job loss was mediated by perceived financial stress. For GAD, there was a significant negative direct effect (*p* = 0.03) and for STB, there was a significant positive total effect (*p* = 0.04). For all other models, the total effect was not significant (all *p* > 0.05).

Mediation analyses with income loss as the independent variable (Figure 3) are displayed in Table 5. Perceived financial stress significantly mediated the relationship between income loss and all mental health conditions, namely depression (*p* < 0.001), GAD (*p* < 0.001), PTSD (*p* < 0.001), panic attacks (*p* < 0.001), SUD (*p* = 0.01), and STB (*p* < 0.001). For these models, the direct effect of income loss on all the mental health conditions was found to be non-significant (all *p* > 0.05). A positive total effect was found for income loss as a predictor of the presence of panic attacks (*p* = 0.02), while for all other models, the total effect was not significant (all *p* > 0.05).

## 4. Discussion

The lockdown in response to the COVID-19 pandemic resulted in sizeable job losses in Spain (and around the world). Indeed, we found that 26.9% of participants considered as active population reported having lost their jobs or being temporarily laid off after the first lockdown during the COVID-19 pandemic, and 42.5% suffered from income loss. Such results are in line with the official records, which showed that Spain’s unemployment increased by 22.9% during 2020, with the highest increase observed in March (9.3%, as compared to February) and April 2020 (8%, as compared to March) [6]. This increment in the rates of unemployment and income loss during the COVID-19 pandemic also seems to be associated with an increased risk of mental health problems, such as depression, anxiety, PTSD, and panic attacks. However, our results indicate that only income loss remained associated with current depression and the presence of panic attacks after adjusting for confounders. Nevertheless, as it was found that losing one’s job and losing income were highly related to each other, this differentiation between income loss and job loss, in terms of their effects on mental health, has to be interpreted with caution.

Our results align with previous findings supporting a relationship between economic hardship and psychological well-being [17,18], as they suggest that losing income during the COVID-19 pandemic was associated with mental health problems (namely, depression and panic attacks). A study conducted in six different European countries also found financial loss to be related to symptoms of depression [25]. Nevertheless, we did not encounter a relationship between job loss and mental health problems directly, as had been found in other studies [21,22,23]. These three studies, conducted in the USA and South Africa, established a direct link between unemployment and higher symptoms of different mental health problems, such as anxiety and stress. According to our results, we could only find significant associations in the non-adjusted models. Whether these national differences might be related to the different policies adopted by governments is still unknown.

For the Spanish economy, the COVID-19 pandemic has taken a great toll after five consecutive years of economic growth [53]. During the COVID-19 outbreak, in March and April 2020, the Spanish government implemented several policies to support the national economy, including aid for small and medium enterprises (SMEs) and self-employed and wage subsidies for employees who had been temporarily laid off [54]. However, when the survey was conducted (in June 2020), there was still uncertainty about the economic situation, and many self-employed individuals, as well as employers of SMEs (which represent 99.8% of enterprises in Spain, [55]), were the hardest hit by the pandemic [56]. Our results suggest that the economic hardship in the context of the COVID-19 outbreak, and especially the prospect of losing income, might negatively impact the mental health of the active population. Thus, labor and social security measures adopted during the pandemic might not only benefit the economic situation of countries but can also help to mitigate the psychological impact. Future studies are needed to evaluate how the implementation of these measures is beneficial for the mental health and wellbeing of the active population.

We hypothesized that the associations between job or income loss and mental health could be mediated by how an individual perceives financial stress [26]. The effect of financial stress on mental health during the COVID-19 pandemic has been found by several studies [30,31,32]. In a study conducted in Australia, financial distress due to the pandemic correlated with worse mental health [30]. Additionally, another recent study conducted in the USA found the relationship between job insecurity and anxiety symptoms to be mediated by financial concern [31]. After conducting mediation analyses, we found perceived financial stress to mediate the relationship between job loss and mental health conditions. The same was reported for income loss, where stress regarding the financial situation significantly mediated the relationship between income loss and all the mental health conditions. Our results are in line with previous studies conducted in the context of other economic crises. These previous findings already pointed towards perceived financial stress as a plausible mediator for the relationship of these variables with mental health [27,28]. A situation of job loss or income loss might lead to this perceived financial stress. Therefore, it is possible that those currently employed during the pandemic, who are experiencing high concerns about their employment status, are reporting higher levels of mental health problems because of their intensified worry about their own financial situation. This stress regarding the own financial situation has been positively associated with depression, anxiety, mood disturbances, burnout, and suicidal ideation in past studies [28]. In our study, when adding perceived financial stress as mediator, the direct relationship between both job loss and income loss with mental health outcomes became non-existent or even turned into a significant negative effect. This is possible since an indirect effect can also exist in the absence of total effects, as a total effect is the sum of many different paths of influence, and not all of them are part of the model [48]. This suggests a mediation where the effects of both job loss and income loss on the mental health variables are mediated by the presence of perceived financial stress. This is the case for the effect of job loss on depression and SUD, and the effect of income loss on all the mental health variables (i.e., depression, GAD, PTSD, panic attacks, SUD, and STB). When a direct effect is found to be negative, and the indirect effect is found to be positive, it creates a competitive partial mediation, since the signs of the indirect and direct paths oppose each other. Such competitive partial mediations may imply a possible “hidden” mediator with opposing signs, as compared to the already identified mediator [57]. This is the case for the effect of job loss on GAD, PTSD, and the presence of panic attacks. In these cases, while job loss has an indirect negative impact on mental health through financial stress, the direct effect of job loss on these mental health variables is found to be positive. Further research is needed to identify if there is, indeed, a hidden mediator between the relationships of job loss and mental health. A possible candidate for such mediator could be social support; losing a job might provoke a supporting reaction from a person’s environment, which, in turn, might lead to less mental health problems. Indeed, previous research has widely shown that social support buffers the effect of unemployment on mental health, supporting this hypothesis [58,59,60].

All in all, these results reflect the potential adverse consequences of the economic situation, due to the COVID-19 pandemic, on the mental health of workers in the Spanish population. Based on these findings, it is important for employers to consider the current situation and implement measure to minimize feelings of uncertainty among their employees. Managers could try to reduce financial concerns through allowing their workers to, for example, telework, even if it is with reduced hours and income, so that the workers do not entirely lose their income. Additionally, to reduce the pandemic impact on mental health of the population, it would be important to increase the access to mental health services in the community. It would be also beneficial for clinicians to screen for job insecurity and provide appropriate referrals to resources, such as unemployment benefits. Lastly, policymakers should consider the long-term effects that may result from employment losses in the community when implementing measures to limit the transmission of COVID-19.

The use of a large community-representative sample of Spanish adults, from a variety of socioeconomic backgrounds, and the ability to control for confounding factors, are strengths of this study. However, several limitations should be considered when interpreting the results. First, the cross-sectional nature of the study does not make it possible to establish the actual directions of the relationships. Additionally, the survey was conducted in a time of high uncertainty, right after the first lockdown. Thus, longitudinal studies are needed to explore how these findings evolve over time. Moreover, since all the answers were self-reported, the results might be affected by recall and reporting biases, which could have led to a social desirability bias, resulting in outcomes that did not fully represent the reality. Moreover, post-stratification weights were based on three sociodemographic variables only, and the survey cooperation rate was low (19.7%), which could have led to a non-response bias. Evidence from previous studies suggests that mental disorders are higher among survey non-respondents [61], suggesting a possible underestimation of mental health problems in our study.

## 5. Conclusions

The COVID-19 outbreak and pandemic, has taken a significant toll on the Spanish economy, resulting in high rates of job loss and income loss in employees. Our results suggest the potential adverse consequences that job and income loss, together with financial concerns, might have on workers’ mental health. These findings highlight the importance of implementing social and financial policies during the COVID-19 pandemic to prevent health inequalities. As such, when implementing measures to limit the transmission of COVID-19, governments should also keep limiting work losses and maintaining financial security among their labor force. Additionally, secondary interventions focused on increasing coping strategies for employees might result in an improvement in their mental health.

## Figures and Tables

**Figure 1 ijerph-19-03158-f001:**
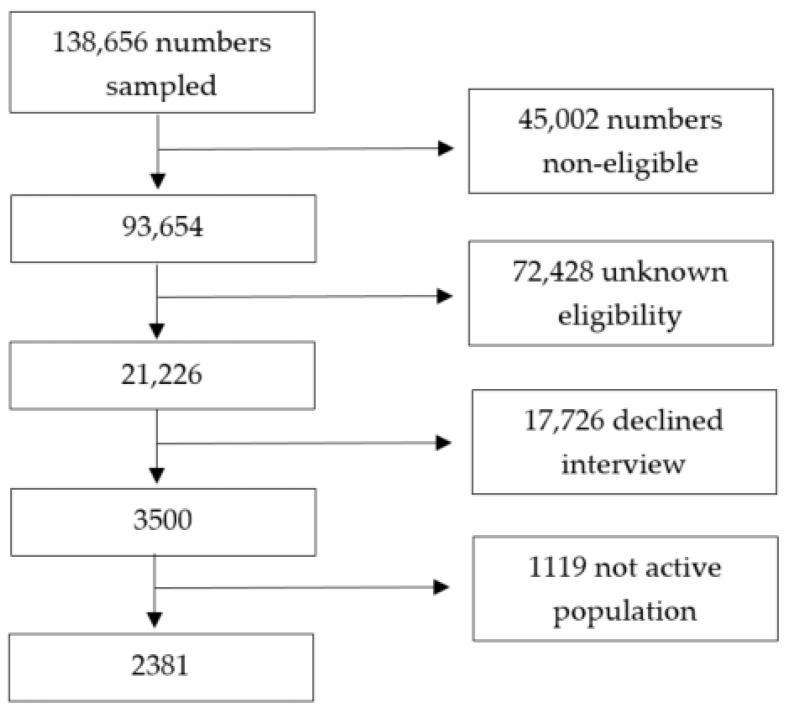
Flow-chart of the study sample.

**Figure 2 ijerph-19-03158-f002:**
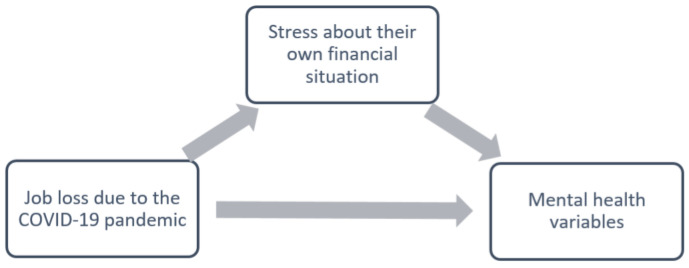
Hypothetical mediation model, with job loss as independent variable.

**Figure 3 ijerph-19-03158-f003:**
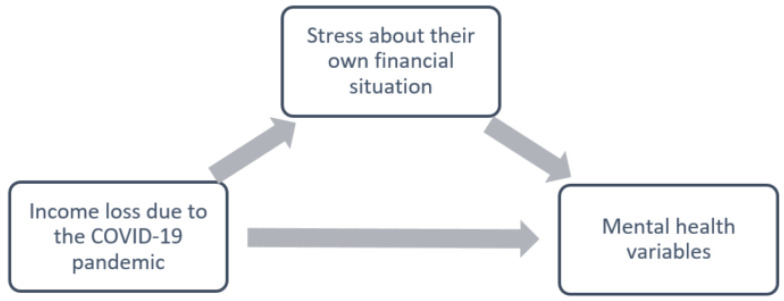
Hypothetical mediation model, with income loss as independent variable.

**Table 1 ijerph-19-03158-t001:** Sociodemographic, mental health, and work-related information in the sample of the active population (*n* = 2381).

	Weighted
	Mean ± SD or % (95% CI)
Age (Mean ± SD)	43.0 ± 11.8
Gender (%)	
Female	47.48% (45.39–49.58%)
Male	52.53% (50.42–54.61%)
Marital Status (%)	
Single	42.38% (40.32–44.47%)
Married	47.87% (45.77–49.96%)
Divorced	8.81% (7.66–10.06%)
Widower	0.94% (0.59–1.44%)
Education Level (%)	
Primary school	2.88% (2.22–3.66%)
Secondary school	16.55% (15.02–18.16%)
High school	13.89% (12.48–15.40%)
Professional Training, intermediate degree	9.26% (8.09–10.54%)
Professional Training, higher degree	11.92% (10.60–13.33%)
Bachelor’s degree	34.12% (32.16–36.14%)
Postgraduate degree	11.38% (10.09–12.77%)
Working situation (%)	
Employed	71.67% (69.75–73.54%)
Employed on sick leave for more than 3 months	3.24% (2.54–4.06%)
Unemployed receiving benefits/subsidy	13.22% (11.84–14.70%)
Unemployed without benefits/subsidy	11.92% (10.60–13.33%)
Depression (% above cut-off)	11.56% (10.26–12.96%)
GAD (% above cut-off)	11.42% (10.13–12.82%)
PTSD (% above cut-off)	10.57% (9.32–11.92%)
Panic attacks (% presence of panic attacks)	10.39% (9.15–11.73%)
SUD (% above cut-off)	3.60% (2.86–4.46%)
STB (% presence of STBs)	3.78% (3.02–4.66%)
Unemployment due to pandemic (%)	
Yes	26.93% (25.10–28.83%)
No	73.25% (71.35–75.08%)
Loss of income due to pandemic (%)	
Yes	42.49% (40.43–44.58%)
No	57.55% (55.47–59.62%)
Perceived financial stress (Mean ± SD)	2.48 ± 1.33

Note. SD = standard deviation; 95% CI = 95% confidence interval; GAD = generalized anxiety disorder; PTSD = post-traumatic stress disorder; STB = suicidal thoughts and behaviors; SUD = substance use disorder.

**Table 2 ijerph-19-03158-t002:** Unadjusted odds ratio (OR) for job and income loss (weighted sample).

	Depression	GAD	PTSD	Panic Attacks	SUD	STB
Job loss	1.48 (1.12–1.95) *	1.19 (0.90–1.59)	1.35 (1.01–1.81) *	1.26 (0.94–1.69)	0.98 (0.59–1.63)	2.11 (1.36–3.28) **
Income loss	1.30 (1.00–1.69) *	1.28 (0.99–1.67)	1.37 (1.04–1.79) *	1.49 (1.13–1.96) *	0.88 (0.55–1.38)	0.98 (0.63–1.52)

Note. N = 2381. * *p* < 0.05, ** *p* < 0.001. CI = confidence interval; GAD = generalized anxiety disorder; PTSD = post-traumatic stress disorder; STB = suicidal thoughts and behaviors; SUD = substance use disorder.

**Table 3 ijerph-19-03158-t003:** Adjusted odds ratio (OR) for job and income loss (weighted sample).

	Depression	GAD	PTSD	Panic Attacks	SUD	STB
Job loss	1.20 (0.90–1.60)	0.93 (0.69–1.26)	1.11 (0.82–1.50)	1.02 (0.75–1.39)	0.80 (0.47–1.34)	1.67 (1.06–2.63) *
Income loss	1.17 (0.90–1.52)	1.14 (0.87–1.49)	1.27 (0.96–1.68)	1.39 (1.05–1.84) *	0.78 (0.49–1.23)	0.85 (0.54–1.33)

Note. N = 2381. * *p* < 0.05. Each model adjusted for gender, age, marital status, educational level. CI = confidence interval; GAD = generalized anxiety disorder; PTSD = post-traumatic stress disorder; STB = suicidal thoughts and behaviors; SUD = substance use disorder.

**Table 4 ijerph-19-03158-t004:** Mediation analyses with job loss as independent variable.

Outcome			Coefficient (95% CI)	*p*-Value
Depression		Total Effect	0.02 (−0.01, 0.05)	0.19
(Yes/No)		Direct Effect	−0.01 (−0.04, 0.02)	0.52
	Financial Stress	Indirect Effect	**0.03 (0.02, 0.04)**	**<0.001 ****
PTSD (Yes/No)		Total Effect	0.01 (−0.02, 0.04)	0.47
		Direct Effect	−0.01 (−0.04, 0.01)	0.31
	Financial Stress	Indirect Effect	**0.02 (0.02, 0.03)**	**<0.001 ****
GAD (Yes/No)		Total Effect	−0.01 (−0.03, 0.02)	0.71
		Direct Effect	**−0.03 (−0.06, 0.00)**	**0.03 ***
	Financial Stress	Indirect Effect	**0.02 (0.02, 0.03)**	**<0.001 ****
Panic attacks		Total Effect	0.00 (−0.02, 0.03)	0.87
(Yes/No)		Direct Effect	−0.02 (−0.04, 0.00)	0.08
	Financial Stress	Indirect Effect	**0.03 (0.02, 0.04)**	**<0.001 ****
SUD		Total Effect	−0.01 (−0.02, 0.01)	0.43
(Yes/No)		Direct Effect	−0.01 (−0.03, 0.00)	0.15
	Financial Stress	Indirect Effect	**0.01 (0.00, 0.01)**	**0.01 ***
STB		Total Effect	**0.02 (0.00, 0.04)**	**0.04 ***
(Yes/No)		Direct Effect	0.01 (−0.00, 0.03)	0.16
	Financial Stress	Indirect Effect	**0.01 (0.00, 0.01)**	**0.003 ***

Note = in bold, significant effect. N = 2381. * *p* < 0.05, ** *p* < 0.001. Adjusted for gender, age, marital status, educational level. 95% CI = 95% confidence interval; GAD = generalized anxiety disorder; PTSD = post-traumatic stress disorder; STB = suicidal thoughts and behaviors; SUD = substance use disorder.

**Table 5 ijerph-19-03158-t005:** Mediation analyses with income loss as independent variable.

Outcome			Coefficient (95% CI)	*p*-Value
Depression		Total Effect	0.02 (−0.01, 0.04)	0.25
(Yes/No)		Direct Effect	−0.02 (−0.05, 0.01)	0.12
	Financial Stress	Indirect Effect	**0.04 (0.03, 0.05)**	**<0.001 ****
PTSD (Yes/No)		Total Effect	0.02 (−0.00, 0.05)	0.10
		Direct Effect	−0.01 (−0.03, 0.02)	0.53
	Financial Stress	Indirect Effect	**0.03 (0.02, 0.04)**	**<0.001 ****
GAD (Yes/No)		Total Effect	0.01 (−0.01, 0.04)	0.32
		Direct Effect	−0.02 (−0.05, 0.01)	0.18
	Financial Stress	Indirect Effect	**0.03 (0.02, 0.04)**	**<0.001 ****
Panic attacks		Total Effect	**0.03 (0.00, 0.06)**	**0.02 ***
(Yes/No)		Direct Effect	−0.00 (−0.03, 0.02)	0.91
	Financial Stress	Indirect Effect	**0.03 (0.02, 0.04)**	**<0.001 ****
SUD		Total Effect	−0.00 (−0.02, 0.01)	0.30
(Yes/No)		Direct Effect	−0.02 (−0.03, 0.00)	0.06
	Financial Stress	Indirect Effect	**0.01 (0.00, 0.01)**	**0.01 ***
STB		Total Effect	−0.01 (−0.02, 0.01)	0.50
(Yes/No)		Direct Effect	−0.02 (−0.03, 0.00)	0.06
	Financial Stress	Indirect Effect	**0.01 (0.00, 0.02)**	**<0.001 ****

Note = in bold, significant effect. N = 2381. * *p* < 0.05, ** *p* < 0.001. Adjusted for gender, age, marital status, educational level. 95% CI = 95% confidence interval; GAD = generalized anxiety disorder; PTSD = post-traumatic stress disorder; STB = suicidal thoughts and behaviors; SUD = substance use disorder.

## Data Availability

The data that support the findings of this study are available from the corresponding author, B.O., upon reasonable request.

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
