# Peer review of "The Mental Health of Employees with Job Loss and Income Loss during the COVID-19 Pandemic: The Mediating Role of Perceived Financial Stress"

_ijerph, 2022, doi:10.3390/ijerph19063158_

Round 1

Reviewer 1 Report

In this text, the authors provide a study of the association of job and income loss with mental health and the possible mediating role of perceived financial stress during the COVID-19 outbreak. Through interviews, they measured the presence of several mental health issues in a population of 2381 Spanish workers. The authors claim that their results indicate that job and income loss play an essential role in raising mental health issues during the pandemic.

The first positive aspect of this work is the interview process with a relevant-sized population. Although they represent a single culture, the number of interviewed workers gives valuable insights into how general populations worldwide could respond to the same stress.

The reviewer's first theoretical question to the authors comes in the methodology in the Statistical Analysis subsection. The authors state that income loss and job loss were independent variables. How so? How did the authors consider they were independent for statistical analysis? From common sense, these variables should be highly correlated. In the third paragraph, the statement makes more sense as the authors suggest using one or the other. Later in the text, it seems like the variables were tested independently as inputs of the logistic regression models to obtain the OR. Please clarify that in the text.

Even if they tested the variables independently, if the authors consider using both variables as independent on analysis, they should provide a variable independence test to guarantee they are not dependent (chi-square, for instance). They cannot be considered independent input variables in any modeling if they are dependent. Furthermore, if they are highly correlated, they could be interpreted as similar inputs for the modeling. In the results, the authors suggest that this is not the case, as they present an early conclusion on page 9 suggesting income loss and job loss have different effects on mental health issues. This affirmation cannot be truly verified without an independence test. Please verify this issue.

Some topical issues on the writing are:

  • The abstract does not present a single line of research context but goes straight into the topic. It would be interesting to have one or two lines in the beginning, explaining the work's context.
  • Citations are not in the format indicated by the journal. They should be between brackets "[]," with the number of the corresponding reference(s) inside. This is important to verify that the authors' references are all cited within the text.
  • Sections and Subsections have no numbering. Also, the authors did not provide any line numbering for the draft. This made the review process more difficult.

Author Response

Comment 1: The reviewer's first theoretical question to the authors comes in the methodology in the Statistical Analysis subsection. The authors state that income loss and job loss were independent variables. How so? How did the authors consider they were independent for statistical analysis? From common sense, these variables should be highly correlated. In the third paragraph, the statement makes more sense as the authors suggest using one or the other. Later in the text, it seems like the variables were tested independently as inputs of the logistic regression models to obtain the OR. Please clarify that in the text.

Even if they tested the variables independently, if the authors consider using both variables as independent on analysis, they should provide a variable independence test to guarantee they are not dependent (chi-square, for instance). They cannot be considered independent input variables in any modeling if they are dependent. Furthermore, if they are highly correlated, they could be interpreted as similar inputs for the modeling. In the results, the authors suggest that this is not the case, as they present an early conclusion on page 9 suggesting income loss and job loss have different effects on mental health issues. This affirmation cannot be truly verified without an independence test. Please verify this issue.

Response 1: Thank you very much for this important comment. Indeed, the variables were tested separately in two different models, meaning that they were each used as independent variable in two different models. For clarification, we rephrased the sentence explaining the unadjusted logistic regression so that it is clearer: “First, two unadjusted logistic regression models were estimated with the mental health conditions (depression, anxiety, PTSD, panic attacks, substance abuse and STB) as dependent variables and job loss and income loss as independent variable in one of the models and income loss in the other ones.” We also included an independence test for the two explanatory variables, resulting in a significant result: “Job loss and income loss were found to be significantly related to one another (χ2 = 222.02, df = 1, p < .001).“ Therefore, they could indeed be interpreted as similar inputs for the modeling. As such, we have highlighted in in the Discussion section the following: “Nevertheless, as it was found that losing one’s job and losing income were highly related to each other, this differentiation between income loss and job loss in terms of their effect on mental health has to be interpreted with caution.”

Comment 2: Some topical issues on the writing are:

  • The abstract does not present a single line of research context but goes straight into the topic. It would be interesting to have one or two lines in the beginning, explaining the work's context.

  • Citations are not in the format indicated by the journal. They should be between brackets "[]," with the number of the corresponding reference(s) inside. This is important to verify that the authors' references are all cited within the text.

  • Sections and Subsections have no numbering. Also, the authors did not provide any line numbering for the draft. This made the review process more difficult.

Response 2: Thank you very much for these comments. We agree with the reviewer 1 that the abstract did lack of some background information. As such, we have included the following two sentences at the beginning of the abstract to give this background: “COVID-19 outbreak, which was followed by home confinement, is expected to have had pro-found negative impact on people’s mental health. Associated factors, such as losing jobs and in-come, can be expected to lead to an increased risk of suffering from psychopathological problems.” As also pointed out by reviewer 1, we have changed the citation format to adhere with the Journal’s preferred format and we have also added numbering to the sections and subsections to make it more easily readable for the users.

Reviewer 2 Report

Dear authors,

congrats on your work. Please find below all my suggestions to improve it.

  1. Originality:

the study supports the need to investigate more in this field. Similar studies have been published to which authors should better refer. It would be helpful for the authors to better describe why this paper is new and what knowledge gap it is filling.

This article needs to be reviewed in each section. It is discouraged to write in the first person (we studied … our sample … we analyzed … and so on…). So, please, change this form in the entire paper.

Even if the abstract is unstructured, it should follow these sections:• backgrounds: the context and purpose of the study; • Methods: how the study was performed and statistical tests used; • Results: the main findings; • Discussion; • Conclusions: brief summary and potential implications. Especially the two last sections need to be developed and strengthened.

  1. Relationship to Literature:

It is poor. Further literature reviews are needed to explain the concept of "pandemic – economic crises relationship". There is what it does (with covid-19), what it produces, but not what it means and what it will be produced in the economic context.

The whole Introduction paragraph needs to be better explained. This would be an academic article, thus, a lot of effort is needed for this. In particular, authors should delve into the literature review on "financial threat", investigate what it means (from an academic point of view), and give a logical and scientific approach to the manuscript as a whole. The latter should be improved in the manuscript.

In addition, articles that have done the same work in an international context should be cited. This would guarantee an international comparison in the same field of study.

Lastly, you asserted: “If perceived financial stress mediates the impact of loss and income job on mental health during the COVID-19, this could help in the design and adoption of interventions for employees to improve their coping strategies and decrease their level of stress, which in turn might result in an improvement in their mental health” What it means? What do the authors want to say? Please, deepen it.

  1. Methodology:

This method section is ok. But replying to the following questions and structuring the methodology in a scientific way could be useful.

How did participants become enrolled in the study?

What are inclusion and exclusion criteria?

Did you use a validated or not questionnaire?

You can delete the paragraph “Ethics statement” and include the statement at the end of the previous paragraph. It is not necessary to create a new paragraph for two/three sentences.

The methodology should aim to clearly show the steps you have taken to achieve your goals. More clarity is required in this section and to achieve it, the section should be revised, streamlined, and simplified. You need to make it more goal-oriented and explain what you did, without adding compound sentences that only lead to confusion.

In general, the whole methodology structure (steps carried out to perform the model) should be better written and reorganized. To do this, I suggest inserting a chronological order of all steps you followed, possibly, adding the answers to the questions abovementioned, and, also, a few hints related to the questionnaire used.

  1. Discussion and Conclusions:

The implications need to be redefined. Restate your topic briefly and explain why it’s important. Make sure that this discussion part is concise and clear. You should have already been clear about why your arguments are important in this part of your paper, and you also don’t need to support your ideas with new arguments. The list of implications should be deepened. This is a scientific article, therefore it needs to be strengthened from the point of view of academic language. Anyway, I would suggest that a discussion section should be developed taking into account your initial research question and a clear statement of proposed contributions, once you have rephrased your arguments and developed some propositions.

In addition, in a discussion, the results obtained in the study are compared with the results obtained by other researchers, or in any case, are commented on. Here there are no comparisons of results. This is a semi quali-quantitative paper, so it should be easy to comment on the results obtained.

Lastly, since the conclusions are too few, I would suggest creating one paragraph called “Discussion and Conclusion”, adding a list of practical, social, and academic implications of your study.

Author Response

Comment 1: 1.           Originality:

the study supports the need to investigate more in this field. Similar studies have been published to which authors should better refer. It would be helpful for the authors to better describe why this paper is new and what knowledge gap it is filling.

This article needs to be reviewed in each section. It is discouraged to write in the first person (we studied … our sample … we analyzed … and so on…). So, please, change this form in the entire paper.

Even if the abstract is unstructured, it should follow these sections:• backgrounds: the context and purpose of the study; • Methods: how the study was performed and statistical tests used; • Results: the main findings; • Discussion; • Conclusions: brief summary and potential implications. Especially the two last sections need to be developed and strengthened.

Response 1: Thank you very much for these comments. As it is true that other articles have been published with similar research questions, we have included more of this literature in the paper. Also, we have added some more information to the aims of our study to make the knowledge gap that we are trying to fill clearer:

“As different countries and regions adopted distinct measures to stop the COVID-19 propagation, the impact of the economic consequences in one country might not reflect the consequences from another. Therefore, the aim of this study was to assess the association of job and income loss with mental health problems in Spanish employees right after the first COVID-9 lockdown. Additionally, we aimed to explore the potential mediating role of the stress provoked by the own financial situation in this relationship (i.e., perceived financial stress). It was expected that job and income loss would be associated with increased mental health problems (depression, anxiety, PTSD, panic attacks, substance abuse and STBs). Moreover, perceived financial stress was expected to mediate these relationships. Specifically, it was expected that job and income loss would lead to an increase of stress regarding the own financial situation, which in turn would result in a higher risk of mental health problems. Having a better understanding on how did the economic consequences of the COVID-19 pandemic affect the Spanish population, could guide employers and policymakers on adopting measures to address the issue.”

Regarding the writing in the first person, more and more journals are encouraging such writing, as it becomes a more readable text for the users. Additionally, going through some of the recent article published by the IJERPH are written in the first person. This is why we are keeping it as it was in this sense.

We agree with the Reviewer 2 that the abstract needed more information. We have improved that by adding both, more context about the study and an additional sentence on the implications of the results found:

“COVID-19 outbreak, which was followed by home confinement, is expected to have had pro-found negative impact on people’s mental health. Associated factors, such as losing jobs and in-come, can be expected to lead to an increased risk of suffering from psychopathological problems. Therefore, this study aimed at studying the association of job and income loss with mental health and the possible mediating role of perceived financial stress during the COVID-19 outbreak. The sample included 2,381 Spanish workers who were interviewed right after the first COVID-19 lockdown. Measures were taken for Generalized Anxiety Disorder (GAD); panic attacks; depression; Posttraumatic Stress Disorder (PSTD); substance abuse; suicidal thoughts and behaviours; working conditions; sociodemographic variables and perceived financial stress. Logistic regression models were calculated; with psychological variables as outcomes and job loss and income loss as predictors. Mediation analyses were performed by adding financial threat as mediator. 19.6% and 33.9% of participants reported having lost their job and income due to the pandemic; respectively. Only income loss was related to a higher risk for suffering from depression and panic attacks. When adding financial stress as a mediator; the indirect effects of job and income loss on the mental health measures were found to be significant; there-fore indicating mediation. These findings pinpoint the vulnerability of this population, and highlight the need for interventional and preventive programs targeting mental health in economic crisis scenarios, like the current one.  They also highlight the importance of implementing social and income policies during the COVID-19 pandemic to prevent mental health problems.”

Comment 2: 2.           Relationship to Literature:

It is poor. Further literature reviews are needed to explain the concept of "pandemic – economic crises relationship". There is what it does (with covid-19), what it produces, but not what it means and what it will be produced in the economic context.

The whole Introduction paragraph needs to be better explained. This would be an academic article, thus, a lot of effort is needed for this. In particular, authors should delve into the literature review on "financial threat", investigate what it means (from an academic point of view), and give a logical and scientific approach to the manuscript as a whole. The latter should be improved in the manuscript.

In addition, articles that have done the same work in an international context should be cited. This would guarantee an international comparison in the same field of study.

Lastly, you asserted: “If perceived financial stress mediates the impact of loss and income job on mental health during the COVID-19, this could help in the design and adoption of interventions for employees to improve their coping strategies and decrease their level of stress, which in turn might result in an improvement in their mental health” What it means? What do the authors want to say? Please, deepen it.

Response 2: Thank you for these very important comments. We agree with Reviewer 2 that the depth of the literature could improve. We have made extensive changes to the introduction section to address this issue. In terms of a better description of the “pandemic – economic crises relationship”, we have included the following:

“Since in this situation the source of such an economic impact it is mostly exogenous to the individual and it is mainly due to the appearance of the COVID-19 pandemic, this offers a unique opportunity to analyze the implications of job and income loss for mental health. Previously, the relationship between unemployment and mental health was more difficult to establish, as there were methodological problems associated with causality. Although there is a large literature which has focused on the effects of employment on mental health [8–10], it has been difficult to establish whether poor mental health affected losing the job or if it is the other way around [11]. According to the International Labour Organization’s press release from March 18, 2020, re-ported 5.3 million jobs will be lost due to the COVID-19 pandemic in the low scenario and 24.7 in the high scenario [12]. In terms of mental health, in the high scenario this would translate into an increase in suicides of about 9570 per year. In the low scenario, the unemployment would be associated with an increase of about 2135 suicides.

From the experience of past economic crises, job insecurity conditions, such as the ones workers are experiencing during the pandemic, is thought to lead to a poorer mental health. Three independent systematic reviews, concluded that the people who became unemployed during a recession and/or experienced financial crises were more likely to suffer from poorer health, increased stress, depression, mental hardship, anxiety, and suicidal behaviors [13–15]. Social security programs have the power to buffer the adverse health impacts that such financial crises can cause. Indeed, a review of studies conducted after the recession of 2008 found that unemployed individuals who resided in countries offering better unemployment benefits reported increased physical and mental health due to greater financial security [16].”

We have also deepen into the meaning of financial threat by including: “Financial threat is defined as a self-reported fearful-anxious uncertainty regarding the own current and future financial situation [29]. Such financial threat has been found to be at its highest in individuals who have experienced economic hard-ship [27].”

We have also included more published articles on the topic from authors that have done similar work in an international context. On the one hand articles that have focused mostly on employment loss:

“Indeed, a study conducted in the United States of America (USA) found that individuals who had lost their job reported higher symptoms of depression, anxiety and stress [22]. The same was found to be true in a study conducted in South Africa [23]. Additionally, in other studies conducted also during this period, financial loss has also been related to higher symptoms of mental health problems, such as anxiety or depression [24,25].”

And on the other hand we have also included more literature on articles that focused on financial threat in relation to the COVID-19 pandemic:

“Indeed, several studies conducted during the COVID-19 pandemic encountered the effect of perceived financial stress on mental health [30–32]. As an example, a study regarding the effect of COVID-19 on mental health in the Australian population showed that the financial distress due to the pandemic, was an important correlate of worse mental health, rather than job loss per se [30]. Additionally, another recent study conducted during the COVID-19 pandemic in the USA found that the relation-ship between job insecurity and anxiety symptoms was mediated by financial concern, again reflecting the importance of the subjective experience of job and income loss [31].”

Lastly, we have deepened regarding what is meant by “If perceived financial stress mediates the impact of loss and income job on mental health during the COVID-19, this could help in the design and adoption of interventions for employees to improve their coping strategies and decrease their level of stress, which in turn might result in an improvement in their mental health”, though adding the following text:

“Some strategies have been proposed to reduce the negative psychological state of workers facing the consequences of COVID-19. For example, companies could arrange professional counselors for advice and consultation, or they could provide employees with the ability to control and arrange their work setting in order to help them reduce the negative pressure caused by job insecurity [33].”

Comment 3: 3.           Methodology:

This method section is ok. But replying to the following questions and structuring the methodology in a scientific way could be useful.

How did participants become enrolled in the study?

What are inclusion and exclusion criteria?

Did you use a validated or not questionnaire?

You can delete the paragraph “Ethics statement” and include the statement at the end of the previous paragraph. It is not necessary to create a new paragraph for two/three sentences.

The methodology should aim to clearly show the steps you have taken to achieve your goals. More clarity is required in this section and to achieve it, the section should be revised, streamlined, and simplified. You need to make it more goal-oriented and explain what you did, without adding compound sentences that only lead to confusion.

In general, the whole methodology structure (steps carried out to perform the model) should be better written and reorganized. To do this, I suggest inserting a chronological order of all steps you followed, possibly, adding the answers to the questions abovementioned, and, also, a few hints related to the questionnaire used.

Response 3: Thank you for this very important comment. We agree that a more structure methodology would be helpful for the reader to understand the procedure of the study. We have therefore modified the text in Sample and study design in the Methods section to make the procedure clearer and we have added a figure with a flow-chart of the enrollment of participants. We have highlighted which participants were included as well as the use of validated questionnaires for measuring mental health. The text now says as follows:

“We analyzed data from a cross-sectional survey conducted in a representative sample of non-institutionalized adults in Spain as part of the MIND/COVID project [33]. The target population of the survey included people who were both aged 18 years or older and had no language barriers.Computer-assisted interviews (CATI) were carried out by a bureau of professional interviewers during June 2020. The sample was drawn through a dual-frame random digit dialing (RDD) telephone survey, and it included both landlines and mobile telephones. The distribution of the interviews was planned according to quotas proportional to the Spanish population in terms of age groups, sex and region of residence [34].

A total of 138,656 numbers were sampled, with a final split of 71% mobile telephones and 29% landline. Of them, 45,002 were classified as non-eligible (i.e. 43,120 non-existing numbers, 984 numbers of enterprises, 444 numbers of persons with Spanish language barriers, 268 fax numbers and 186 numbers belonging to quota that were already completed) and 72,428 had unknown eligibility (i.e. no contact was made after the seven attempted calls), resulting in a 19.7% cooperation rate (i.e., the proportion of all cases interviewed of all eligible units ever contacted). Finally, 3500 people were interviewed during the COVID-19 lockdown in Spain, from which only 2381 belonged to the active population (see Figure 1). Interviews included general demographic questions and questions related to social networks and living situation, socioeconomic factors, use of health resources, mental health, general health, and wellbeing. For mental health, depression, anxiety, PTSD symptoms, panic attacks, substance abuse and suicidal thoughts and behaviours were asked by means of validated questionnaires.“

We have also deleted the section title of „Ethics Statement“, and we have just included this section at the end of the Sampe and study design section.

Comment 4: 4.           Discussion and Conclusions:

The implications need to be redefined. Restate your topic briefly and explain why it’s important. Make sure that this discussion part is concise and clear. You should have already been clear about why your arguments are important in this part of your paper, and you also don’t need to support your ideas with new arguments. The list of implications should be deepened. This is a scientific article, therefore it needs to be strengthened from the point of view of academic language. Anyway, I would suggest that a discussion section should be developed taking into account your initial research question and a clear statement of proposed contributions, once you have rephrased your arguments and developed some propositions.

In addition, in a discussion, the results obtained in the study are compared with the results obtained by other researchers, or in any case, are commented on. Here there are no comparisons of results. This is a semi quali-quantitative paper, so it should be easy to comment on the results obtained.

Lastly, since the conclusions are too few, I would suggest creating one paragraph called “Discussion and Conclusion”, adding a list of practical, social, and academic implications of your study.

Response 4: Thank you for all these comments made regarding the discussion. We agree with Reviewer 2 that more implications should have been mentioned and deepened into. Therefore we added a paragraph focusing mostly on possible implications of the results found:

“All in all, these results reflect the potential adverse consequences of the economic situation we are living due to the COVID-19 pandemic on the mental health of workers in the Spanish population. Based on these findings, take the current situation into ac-count and try to minimize feelings of uncertainty in employees should particularly important for the employers to have in mind. Managers could try to reduce financial concerns through allowing their workers to, for example, telework, even if it is with reduced hours and income, so that the workers do no lose entirely their income. Additionally, to reduce the pandemic-related impacts on mental health of the population, it would be important to increase the access to mental health services in the community. Once access is obtained to mental health services, it would be also beneficial for clinicians to screen for job insecurity and provide appropriate referrals to resources, such as unemployment benefits. Lastly, policymakers should consider the long-term effects that may result from employment losses in the community when implementing measures to limit transmission of the COVID-19.

Additionally, throughout the discussion section, we have included more studies that were conducted with similar results and we have made more comparisons between our findings and theirs:

“Our results align with previous findings supporting a relationship between economic hardship and psychological well-being [17,18], as they suggest that losing income during the COVID-19 pandemic was associated with mental health problems (depression and panic attacks). A study con-ducted in six different European countries did also find financial loss to be related to symptoms of depression [25]. Nevertheless, we did not encounter a relationship be-tween job loss and mental health problems directly, as had been found in other studies [21–23]. These three studies, conducted in the USA and South Africa, established a direct link between unemployment and higher symptoms of different mental health pathologies, such as anxiety and stress. According to our results, we could only find such an association when not adjusting for confounders. It remains the question whether this might be related to different policies established by the different governments.“

“We had hypothesized that the associations between job or income loss and mental health could be mediated by how an individual perceives a financial stress [26]. The effect of financial stress on mental health during the COVID-19 pandemic has been found by several studies [30–32]. In a study conducted in Australia financial distress due to the pandemic correlated with worse mental health [30]. Additionally, another recent study conducted in the USA found the relationship between job insecurity and anxiety symptoms to be mediated by financial concern [31]. After conducting media-tion analyses, we encountered perceived financial stress to mediate the relationship between job loss and mental health conditions. The same was found for in-come loss, where stress regarding the own financial situation significantly mediated the relationship between income loss and all the mental health conditions. Our results agree with published articles conducted in the context of other economic crises. These previous findings already pointed towards perceived financial stress to be a plausible mediator for the relationship of these variables with mental health [27,28].”

Reviewer 3 Report

Dear Authors, 

I find your research very topical. And well written. Actually, it is difficult to find important flaws or recommend necessary corrections, in my opinion.

The literature review is sufficient. The research methods are correct, and the results are well described. Alternatively, I can suggest extending the discussion.

Author Response

Comment 1: I find your research very topical. And well written. Actually, it is difficult to find important flaws or recommend necessary corrections, in my opinion.

The literature review is sufficient. The research methods are correct, and the results are well described. Alternatively, I can suggest extending the discussion.

Response 1: Thank you very much for these very appreciating comments

Round 2

Reviewer 1 Report

The authors provided the requested changes, leading to a significant improvement of this text. They should assess some formatting issues leading to the final version. Thus, I recommend accepting this paper in the present format, considering that these corrections will be provided in the following stages.

Reviewer 2 Report

Dear Authors,

by following all of my suggestions, the manuscript has been improved as a whole. Thus, the study is ready for publication.